# Carbon Monoxide or Ruthenium: Will the Real Modulator of Coagulation and Fibrinolysis Please Stand Up!

**DOI:** 10.3390/ijms26083567

**Published:** 2025-04-10

**Authors:** Vance G. Nielsen, Anthony R. Abeyta

**Affiliations:** 1Department of Anesthesiology, The University of Arizona College of Medicine, Tucson, AZ 85724, USA; 2Ross University School of Medicine, Bridgetown BB11037, Barbados; anthonyabeyta@mail.rossmed.edu

**Keywords:** carbon monoxide, ruthenium, heme, histidine, carbon monoxide releasing molecule-2, thrombelastography, fibrinogen, plasmin, α_2_-antiplasmin

## Abstract

The discovery of carbon monoxide releasing molecules (CORMs) was one of the most impactful innovations in biochemistry, affecting multiple disciplines for the past few decades. Sixteen years ago, a ruthenium dimer-containing CORM, CORM-2, enhanced coagulation and diminished fibrinolysis in human plasma by modulation of fibrinogen, plasmin, and α_2_-antiplasmin via CO binding to putative heme groups attached to these proteins. This finding linked CO exposure in settings involving heme oxygenase-1 upregulation during inflammation or environmental exposure to thromboembolic disease in hundreds of subsequent manuscripts. However, CO-independent effects of CORM-2 involving a putative ruthenium radical (Ru•) formed during CO release was found to be responsible for many of effects by CORM-2 in other works. Using a novel approach with human plasmatic coagulation kinetic methods, Ru• was posited to bind to critical histidines and other amino acids to modulate function, and excess histidine to quench CORM-2-mediated effects. This paradigm of histidine addition would definitively address if CO or Ru• was responsible for CORM-2-mediated effects. Thus, plasma coagulation/fibrinolytic kinetic data were assessed via thrombelastography ±CORM-2, ±histidine added. Histidine nearly completely abrogated CORM-2-mediated hypercoagulation in a concentration-dependent fashion; further, histidine also nearly eliminated all kinetic effects on fibrinolysis. In conclusion, CORM-2 Ru• formation, not CO release, is the true molecular mechanism modulating coagulation and fibrinolysis.

## 1. Introduction

The creation and utilization of carbon monoxide releasing molecules (CORMs) was one of the most impactful innovations in biochemistry, affecting multiple disciplines for the past few decades [1,2,3,4,5,6,7,8,9,10]. The ability to release carbon monoxide (CO) in a site-directed manner on the molecular level allowed hundreds of investigations to be conducted to ask and answer questions concerning the role of CO in biochemical reactions, cellular functions, and whole animal responses when administered systemically [1,2,3,4,5,6,7,8,9,10]. Among the earliest demonstrations of a CORM involved a ruthenium (Ru) iron hybrid of a modified heme molecule [1]. Many years later, the earliest descriptions of a dimeric Ru CORM (CORM-2 [tricarbonyldichlororuthenium (II) dimer]) [5] and a water soluble, monomeric Ru CORM (CORM-3 [tricarbonylchloro(glycinato)ruthenium(II)]) [6] that released CO at physiological pH were found to perform as expected in a variety of systems. Six years later in 2009, the senior author of the present work made the seminal finding that CORM-2 enhanced human plasmatic coagulation [7] and diminished fibrinolytic vulnerability to tissue-type plasminogen activator (tPA) [8] in vitro. An experimental pillar to demonstrate that CO, rather than a CORM-2 byproduct produced after the release of CO, was responsible for the phenomena observed was the use of “inactivated” CORM-2 as a negative control that was generated by using a mixture of CORM-2 dissolved in dimethyl sulfoxide (DMSO) that was left at room temperature for a few days until presumably all CO was released as the solution lost its yellow color [7,8]. Plasma exposed to CORM-2 not only had greater velocity of clot growth and final viscoelastic strength, but also had electron microscopic features that included far more fine fibrin polymer formation that was dense [9,10]. These findings were subsequently translated into a sedated rabbit model, with CORM-2 (10 mg/kg) administration intravenously decreasing bleeding time following the administration of clopidogrel (20 mg/kg) with aspirin (10 mg/kg) via gavage [11] or the administration of tPA (1 mg/kg) intravenously [12]. In addition to using inactivated CORM-2 (iCORM-2), the senior author included a variety of experimental verifications of the CO-based modulation of coagulation and fibrinolysis. These studies resulted in a putative molecular mechanism for these CORM-2/CO-mediated phenomena that was based on the modulation of key molecules in the coagulation and fibrinolysis pathways via a cryptic heme attached to them [13,14,15,16]. Specifically, fibrinogen, plasmin, and α_2_-antiplasmin were found to have heme group(s) via mass spectroscopy and/or have modified performance as a substrate or enzyme after exposure to nitric oxide (NO) donors which would be expected to interact with heme groups [13,14,15,16,17]. Fibrinogen became a superior substrate for thrombin [13], plasmin lost activity [14], and α_2_-antiplasmin [14,15] demonstrated increased affinity after being exposed to CORM-2/CO in isolation. These fundamental, mechanistic studies were translated into multiple observational, clinical studies that linked exogenous CO exposure (e.g., smoking) and endogenous CO production (e.g., generated by heme oxygenase in states of inflammation) to hypercoagulable/hypofibrinolytic states that were conducted over several years by the senior author. The basic science and clinical studies conducted by the senior author and the number of citations of these works without inclusion of self-citations are listed in Table 1. The citation data were collected from Google Scholar (https://scholar.google.com/citations, accessed on 26 December 2024).

The paradigm of cryptic heme groups that was responsible for CO-mediated hypercoagulation and hypofibrinolysis seemed well grounded has been accepted up to present day, but there was at least one inconsistency that was first generated by the author. In a work published in 2014 that compared the procoagulant effects of CORM-2, CORM-3, and CORM-A1 (sodium boranocarbonate), it was found that CORM-A1 had no effect on coagulation kinetics but did cause significant hypofibrinolytic effects [18]. It was postulated that boron could stabilize heme bound to fibrinogen, inhibiting the effect of CO on fibrinogen [19], and the inhibition of fibrinolysis by CORM-A1 was consistent with the effect seen with CORM-2 [8]. However, three years later in a separate field, CO-independent inhibition of K^+^ channels with a putative Ru-based radical (Ru•) formed from CORM-2 during CO release was demonstrated [20]. The investigators used an excess of free histidine (1 mM) or albumin (which is resplendent in histidine content [21]) to quench the proposed Ru• formed from CORM-2 [20]. Of interest, the senior author used the same approach, using albumin, to demonstrate that the Ru•, not CO, derived from CORM-2 was an antivenom against honey bee phospholipase A_2_ [22]. As recently demonstrated and reviewed, multiple Ru ionic and radical species interact not just with histidine residues, but also with sulfur-containing amino acids that may be critical to enzymatic activity [23,24]. Given these data, it may be possible that CO–heme interactions were not at play in the settings of coagulation and fibrinolysis (Table 1), and this issue needed to be investigated.

Not surprisingly, histidine and sulfur-containing amino acids are found in fibrinogen [25,26], plasmin [27,28], and α_2_-antiplasmin [29]. Thus, these three key coagulation/fibrinolysis system molecules likely have heme groups that would be targeted by CO or NO, while also having key histidine and other amino acid targets for a CORM-2 derived Ru•. CO and Ru• could modulate these three proteins independently or synergistically, or competitively, etc. While albumin has numerous histidines (16 total), it is also capable of binding heme [30], which could confound separating CO- and Ru•-mediated effects on coagulation and fibrinolysis. Further, if the concentration of albumin needs to be increased experimentally, this can compromise coagulation [31]. Free heme also compromises plasmatic coagulation [32,33], preventing its use to exclude CO-mediated effects. Thus, the most viable approach to separating CO- and Ru•-mediated effects in a plasmatic system would be to increase the concentration of free histidine without compromising coagulation, which would bias towards revealing CO-mediated events by maximizing Ru•–histidine binding and abrogating Ru•-mediated effects.

In addition to using histidine to quench potential Ru• formation when investigating the effects of CORM-2, the utilization of thrombelastographic models of coagulation and fibrinolysis reveal the effects of CORM-2 with thrombin–fibrinogen–factor XIII (FXIII) interactions [7,13] and fibrin polymers (polymerized by thrombin and crosslinked by FXIII)–plasmin–α_2_-antiplasmin interactions [8,14,15], respectively. It is critical to note that the addition of tPA before the commencement of coagulation results in a model of simultaneous clot formation and destruction, as tPA binds to lysine residues that appear on forming fibrin polymers, followed by the rapid conversion of plasminogen to plasmin, which in turn results in further catalysis of fibrin polymers and exposure of more lysine residues [34]. Thus, while coagulation and fibrinolysis can be discussed as separate and sequential biochemical pathways when considering establishing thrombi in vivo followed by thrombolysis, the models utilizing thrombelastography to assess coagulation and fibrinolysis with samples containing tPA have two simultaneous and competing processes contributing to the data collected.

The clinical implications of the potential findings of this study are wide sweeping. As noted in Table 1, hundreds of investigations linking CO to thrombotic phenomena used data generated with CORM-2 as the foundation and rationale for their work. If CO is not responsible and Ru• formation is responsible for the observed changes in coagulation and fibrinolysis associated with CORM-2 exposure, then the association of CO with clinical thrombosis may well be “true, true, but unrelated”. The “house of cards” supporting the link between CO exposure and thrombosis would collapse. Thus, the conduct of the present study was considered critical to elucidate the true mechanism of a CORM-2-mediated enhancement of the coagulation and attenuation of fibrinolysis in human plasma.

Given the previously presented information, the goals of the present study were as follows. First, to address a practical matter, a comparison of the change in clot growth parameters in pooled normal human plasma without or with tPA or CORM-2 was to be performed to establish the degree of clot lysis during clot growth in the presence of tPA or CORM-2. Second, the effects of the addition of free histidine to human plasma on coagulation kinetics in the absence of CORM-2 or prior to CORM-2 addition was to be accomplished to determine if histidine by itself interferes with coagulation and if the amino acid changes CORM-2-mediated effects. Third, plasma pretreated with histidine addition in the absence or presence of CORM-2 was to have tPA addition to determine if not just coagulation but fibrinolytic clot kinetics are modified by the potential quenching of a Ru•. Fourth, to assess the reversibility of CORM-2 effects, histidine was to be added to plasma already exposed to CORM-2 and tPA with coagulation and fibrinolytic parameters assessed. Figure 1 displays the relevant coagulation and fibrinolytic enzymes affected by CORM-2 and NO donors, and the corresponding thrombelastographic parameters associated with the processes of thrombus growth and dissolution are also presented.

## 2. Results

### 2.1. Comparison of Coagulation and Fibrinolysis Kinetics of Human Plasma Without or with tPA and Without or with CORM-2

The kinetic behavior of the lot of pooled normal human plasma activated by tissue factor needed to be characterized in the absence and presence of both tPA and CORM-2 prior to experimentation with additions of histidine. To that end, equal volumes of human plasma were exposed to vehicle control (DMSO) or CORM-2 (100 µM final concentration) with the addition of vehicle control (dH_2_O) or tPA (500 U/mL or 0.86 µg/mL). Both vehicle/compound additions were 1% (*v*/*v*) in plasma, with further addition of a 3% addition of dH_2_O in anticipation of comparison with plasma exposed to histidine as presented in Section 2.2. It should be noted that the concentration of tPA was five-fold greater than what has been utilized in the past [8,12,14,15], with the rationale that a greatly increased fibrinolytic stress will result in a more rapid completion of assays that could better define the effects of CORM-2 and histidine during fibrinolysis. The results of these experiments are displayed in Figure 2 and Table 2.

The analyses of the coagulation kinetic data are as follows. As noted previously in [8,12,14,15], neither the addition of CORM-2 nor tPA significantly affected TMRTG. In sharp contrast, the MRTG and TTG values of all four conditions were significantly different from each other. The addition of CORM-2 increased MRTG (102%) and TTG (58%) values compared to the control condition; further, the addition of tPA decreased MRTG (13%) and TTG (43%) compared to the control condition. Compared to the CORM-2 addition condition, the addition of tPA to CORM-2 exposed samples resulted in a decrease in MRTG (5%) and TTG (21%). The enhancement of coagulation kinetic values by CORM-2 and degradation by tPA was consistent with previous works utilizing 100 µM CORM-2 with or without 100 U/mL tPA [7,8,12,13,14,15].

The fibrinolytic kinetic data analyses demonstrated a significant increase in CGT (62%), TMRL (183%), MRL (34%), and CLT (118%) in the condition with both CORM-2 and tPA addition compared to the condition of tPA addition alone. While these data demonstrate for the most part an antifibrinolytic effect of CORM-2 consistent with previous work [8,12,14,15], the MRL has previously been decreased, not increased, in the presence of CORM-2. We speculate that the most likely molecular explanation for the increase in MRL observed following the addition of 500 U/mL tPA compared to the decrease in MRL following the addition of 100 U/mL of tPA in the presence of CORM-2 is that the distance between lysine residues exposed on newly formed fibrin polymers is far smaller than that observed in the absence of CORM-2 [9,10]. As revealed by both transmission and scanning electron microscopy, thrombi exposed to 100 µM CORM-2 form dense, thin fibrin polymer meshes [9,10] that would allow more frequent interactions of the increased tPA concentration in the present study with lysine residues closely approximated within the thrombus. Nevertheless, the antifibrinolytic effects of CORM-2 on the activities of plasmin and α_2_-antiplasmin manifested by increased CGT, TMRL, and CLT overshadowed the effects on MRL. Given the significant effects of CORM-2 on fibrinolytic kinetics at this concentration of tPA, it was deemed acceptable to proceed with subsequent experimentation with histidine addition as planned.

### 2.2. Coagulation Kinetics of Human Plasma with Increasing Concentrations of Histidine

The rationale for this series of experiments was that while a concentration of 1 mM histidine was used to demonstrate CO-independent effects of CORM-2 [20], these experiments were conducted in a buffer milieu. As already mentioned, human plasma has many proteins with multiple histidines and disulfide bonds [25,26,27,28,29,30]; for example, albumin is present at a concentration of approximately 640 µM [35] and contains 16 histidines [21], presenting 10.2 mM of histidine by itself. The concentration of free histidine in plasma is 70–120 µM [36]; in contrast, the plasma concentrations of the target molecules for CORM-2-derived products are 6–12 µM for fibrinogen [37], 2 µM for plasminogen [38], and 1 µM for α_2_-antiplasmin [39]. As 100 µM CORM-2 only releases 70 µM CO [5], and presumably also only forms 70 µM of Ru•, it is remarkable that fibrinogen, plasmin, and α_2_-antiplasmin with their small plasmatic concentrations compared to the vast mM concentrations of histidine and other molecular targets could bind CORM-2 products. It is therefore possible that the affinity of potential Ru• for function-critical histidines in fibrinogen, plasmin, and α_2_-antiplasmin may be greater than for free histidine. Thus, the concentrations of histidine were tested (Figure 3), with the limitation of 8mM accepted to maintain a dilution of only 3% of plasma given the solubility of histidine in water used as the vehicle (41.6 g/L; histidine 155.15 gm/mol molecular weight). The interested reader will find information, in greater detail, concerning the interactions of CORM-2-derived Ru• with amino acids besides histidine and with disulfide bridges as recently reviewed [23]. In summary, this range of histidine concentrations would not be expected to affect plasmatic coagulation kinetics. The results of this series of experiments are displayed in Figure 3. Human plasma exposed to the range of histidine demonstrated no significant changes in TMRTG, MRTG, or TTG between the conditions.

### 2.3. Coagulation Kinetics of Human Plasma with Increasing Concentrations of Histidine Prior to Addition of CORM-2

The data generated from these experiments are depicted in Figure 4. Histidine decreased CORM-2-mediated increases in MRTG and TTG in a concentration-dependent fashion. While 8 mM histidine in the presence of CORM-2 did not reduce MRTG values to that of plasma exposed only to 8 mM histidine, the difference between the conditions was only 12%. In contrast, TTG values of the condition of 8mM histidine alone and 8 mM histidine followed by CORM-2 exposure were not significantly different. Thus, these data strongly support the concept that fibrinogen-dependent enhancement of plasmatic coagulation by CORM-2 is achieved with a Ru•-dependent mechanism. Given the results of this series of experiments, 8 mM histidine was utilized for all subsequently described experimentation.

### 2.4. Comparison of Coagulation and Fibrinolysis Kinetics of Human Plasma with tPA, Without or with CORM-2, Without or with Histidine

The coagulation kinetic data and fibrinolytic kinetic data generated from these experiments are displayed in Figure 5 and Figure 6, respectively. As seen in Figure 5, histidine addition did not significantly affect TMRTG, MRTG, or TTG values compared to the condition with tPA addition alone. However, the addition of histidine prior to the addition of CORM-2 resulted in MRTG and TTG values significantly lower than samples exposed to CORM-2 but not different from samples exposed to histidine alone. However, compared to samples with tPA addition alone, samples exposed to histidine, CORM-2 and tPA had significantly greater MRTG and TTG values. Nevertheless, given the lack of significant difference between the conditions with histidine addition, it appears that coagulation kinetic changes caused by CORM-2 may be largely attributed to a Ru•-dependent mechanism. The data in Figure 6 demonstrate that histidine addition did not significantly affect CGT, CLT, TMRL, or MRL values compared to the condition with tPA addition alone, indicating that the process of fibrinolysis induced by tPA was not affected by 8 mM histidine just as coagulation was not affected by histidine in the absence of tPA (Figure 3). Critically, histidine addition prior to CORM-2 addition significantly decreased CGT, decreased CLT, decreased TMRL, and decreased MRL compared to samples with only tPA and CORM-2 addition. There were no significant differences in CGT, CLT, TMRL, or MRL values between the samples exposed to tPA and histidine compared to the condition with histidine, CORM-2, and tPA addition. However, the TMRL values derived from the group with tPA, histidine, and CORM-2 addition were significantly greater than the condition with tPA addition alone. Be that as it may, the overall pattern of kinetic change favors a significant reduction in CORM-2-mediated effects on fibrinolysis via a putative Ru•-dependent mechanism.

In summary, the vast majority of changes in plasmatic coagulation and fibrinolytic kinetics caused by CORM-2 addition appear to be secondary not to a CO-dependent mechanism but instead a CO-independent mechanism involving a putative Ru•, the effects of which are quenched by histidine.

### 2.5. Determination of Reversibility of CORM-2-Mediated Effects on Coagulation and Fibrinolysis Kinetics of Human Plasma with tPA by Exposing Plasma to CORM-2 Prior to Histidine Addition

The data generated by this final series of experiments are depicted in Figure 7 and Figure 8. The primary rationale for these experiments was to determine if the potential Ru• bound irreversibly to target amino acid residues, given that radicals tend to have stronger bonds with ligands than simple ionic or covalent bonding. However, if the reversibility of CORM-2 effects on coagulation and fibrinolytic kinetics could be demonstrated, this finding would still implicate the interaction of a Ru molecular moiety with target proteins that would involve a CO-independent mechanism.

Regarding plasmatic coagulation kinetic data displayed in Figure 7, the addition of histidine after CORM-2 addition did not significantly affect TMRTG values compared to the two other conditions. Further, MRTG values were significantly greater in both conditions with CORM-2 addition compared to samples with tPA alone; however, samples exposed to histidine after CORM-2 addition had significantly smaller MRTG values. Lastly, the two conditions with CORM-2 addition resulted in significantly greater TTG values compared to samples with tPA addition alone, and the addition of histidine after CORM-2 addition did not significantly change TTG values compared to the condition with CORM-2 addition before tPA addition.

As for the plasmatic fibrinolytic kinetic data displayed in Figure 8, CGT was significantly increased in the two conditions with CORM-2 addition compared to the samples with tPA addition alone, and the addition of histidine after the addition of CORM-2 did not significantly modify CGT. However, while both CORM-2 addition conditions had significantly greater CLT compared to samples with tPA alone, the condition with histidine addition after CORM-2 addition had significantly smaller CLT values compared to samples with CORM-2 addition alone. Like CLT values, TMRL values in both CORM-2 addition conditions were significantly greater than samples with tPA alone, and the condition with histidine addition after CORM-2 addition had significantly smaller TMRL values compared to samples with CORM-2 addition alone. Lastly, MRL values were significantly greater in the two conditions with CORM-2 addition compared to the condition with tPA addition alone; however, the condition with histidine addition after CORM-2 addition had larger MRL values than the condition with CORM-2 addition alone.

When these data are considered together, it seems that the reversibility of CORM-2-mediated effects on plasmatic coagulation kinetics by histidine is very minor at best given the small significant decrease in MRTG values and no significant decrease in TTG values in the condition with histidine addition before CORM-2 compared to the samples with CORM-2 addition alone. In contrast, histidine addition after CORM-2 addition was found to partially reverse CORM-2-mediated effects on plasmatic fibrinolytic kinetics by significantly decreasing CLT values and decreasing TMRL values. The significant increase in MRL values by histidine addition after CORM-2 addition compared to the condition of CORM-2 addition alone is difficult to precisely explain, but it could be speculated that plasmin activity could be enhanced via the displacement of a Ru• by histidine and the CORM-2-mediated formation of dense polymer fibers would allow enhanced proteolysis. A complementary mechanism could be that α_2_-antiplasmin activity could be decreased by the displacement of a Ru• by histidine, decreasing the rate of the irreversible removal of plasmin. Lastly, it could be a combination of both these mechanisms that partially diminish the antifibrinolytic effects of CORM-2. In summary, it appears that CORM-2-mediated enhancement of plasmatic coagulation kinetic data associated with a putative Ru•-based mechanism is not as easily reversed by histidine as is the plasmatic fibrinolytic kinetic data associated with CORM-2 addition.

## 3. Discussion

This investigation achieved its four goals, providing data that easily build towards identifying potential Ru•-mediated effects on plasmatic coagulation and fibrinolytic kinetics during CORM-2 exposure. First, the kinetic behavior of plasmatic thrombus formation in the absence of tPA compared to the model of simultaneous clot growth and destruction in the presence of tPA needed to be performed to document expected changes in thrombelastographic parameters prior to experimentation with histidine and CORM-2. Next, demonstrating that at large concentrations of histidine with minimal dilution of plasma by histidine additions there were no significant effects on typical plasmatic coagulation or fibrinolytic kinetics was critical. Furthermore, the observation of a concentration dependent decrease in the CORM-2 enhancement of coagulation by progressively greater histidine concentrations served as evidence of a primary Ru•-mediated mechanism at play, not a CO-mediated mechanism. In a complementary fashion, the marked attenuation of CORM-2-mediated effects on plasmatic fibrinolytic kinetic data strongly supported a Ru•-mediated and not CO-mediated mechanism as key. Lastly, while histidine addition after CORM-2 exposure seemed to not remarkably reverse the enhancement of coagulation kinetic data, the significant, partial reversibility of CORM-2-mediated decreases in fibrinolytic kinetic data demonstrate a potentially smaller affinity of Ru• for molecular targets in the fibrinolytic pathways compared to fibrinogen in the coagulation pathway. When considered as a whole, the data of the present investigation strongly support a mechanism involving Ru• generated from CORM-2 during CO release as responsible for the modulation of coagulation and fibrinolysis following CORM-2 exposure.

While the mechanism of the modulation of plasmatic coagulation and fibrinolysis by CORM-2 may be unambiguous, it may be premature to assert that the heme groups identified on fibrinogen, plasmin, and α_2_-antiplasmin via mass spectroscopy and/or by responsiveness in the presence of NO donors [13,14,15,16,17] play no role in coagulation and fibrinolysis. While CO released from the boron-containing CORM-A1 did not affect plasmatic coagulation kinetic data, CORM-A1 exposure did diminish plasmatic fibrinolytic kinetic data in a manner like CORM-2 [18], leaving the possibility that either CO or a yet to be described boron radical was responsible for these effects. Another possibility is that perhaps the putative heme groups on fibrinogen, plasmin, and α_2_-antiplasmin are primarily responsive to NO modulation, not CO modulation. Critically, in the multiple clinical investigations contained in Table 1, disease conditions involving heme oxygenase-1 generated CO as detected by increased carboxyhemoglobin concentrations were found to have plasmatic hypercoagulation that could not be enhanced to the same degree by exposure of the plasma to CORM-2 compared to normal subjects’ plasma. This assay used to implicate CO as a modulator of coagulation based on CORM-2 exposure was first described as cited [13]. Thus, if documented systemic excess CO generation by heme oxygenase-1 did not play a role in the hypercoagulation caused by diverse inflammatory states (e.g., cancer, obesity, dialysis, migraine headache), then some other common, pathological biochemical modification of fibrinogen must have occurred that rendered fibrinogen no longer responsive to CORM-2 modulation. In conclusion, while the mechanism responsible for modulation of plasmatic coagulation and fibrinolysis by CORM-2 may be defined, the mechanism(s) responsible for hypercoagulation and hypofibrinolysis associated with multiple inflammatory states that may or may not involve the heme groups associated with fibrinogen, plasmin, and α_2_-antiplasmin require future investigation to identify.

One issue that other investigations utilizing free histidine to demonstrate CO-independent, Ru•-dependent effects of CORM-2 is the matter of affinity of the presumed Ru• for amino acids in the protein of interest. CORM-2 at 100 µM presumably generates Ru• at 70 µM in a diffuse manner, interacting with all potential ligands in the immediate microenvironment with a half-time of 1 min [5]. Therefore, within 5 min following CORM-2 addition, it would be expected that essentially all Ru• would be generated and would also likely bind to nearby ligands. So, an expected concentration of 6–12 µM fibrinogen [37], 2 µM plasminogen [38], and 1 µM α_2_-antiplasmin [39], for a total of as much as 15 µM protein target, is significantly activated/inactivated in plasma with 70 µM. As mentioned in Section 2.2 previously, between the histidines contained in albumin or in free solution, over 10 mM histidine is present in plasma, well over the 100-fold the Ru• generated by 100 µM CORM-2. Then, an additional 8 mM free histidine concentration must be added to nearly abrogate the effects of 100 µM CORM-2 on coagulation and fibrinolysis. In total, somewhere between 200- and 300-fold histidine is needed to eliminate the effects of the small concentration of Ru• from interacting with the three key proteins involved in coagulation and fibrinolysis. Thus, it seems that Ru• has a large affinity for fibrinogen, plasmin, and α_2_-antiplasmin compared to other plasmatic proteins or histidine. Further, it is likely that perhaps only a small (30–60%) critical amount of fibrinogen, plasmin, and α_2_-antiplasmin need to be bound by Ru• to achieve the change in plasmatic coagulation and fibrinolytic kinetic data observed thrombelastographically. In summary, the molecular interactions of CORM-2-generated Ru• with key target molecules are complex and may be the focus of future investigation.

The present investigation has several issues to consider. First, plasmatic biochemical events, not cellular matters, were the focus. This has been a limitation for all the senior author’s investigations with CORM-2 over the years, as the interest was on the plasmatic events that could lead to clinical thrombotic disease. Another limitation is that further biochemical validation experiments could be performed, such as spectroscopic analyses or mass spectrometry, to confirm direct binding interactions between Ru• and target coagulation/fibrinolysis proteins (fibrinogen, plasmin, or α_2_-antiplasmin). While such investigation is potentially of interest, it is beyond the scope of the present investigation that had very specific objectives using widely validated methods. A further limitation is the use of our methods beyond the plasmatic biochemical realm. The concentrations of CORM-2, histidine, or any other compounds used in our plasma-based system cannot be adopted into cellular or in vivo systems without requisite concentration–response relationships being defined for the molecular targets of interest and cellular/whole animal toxicity. Another issue is the future clinical directions to pursue considering the present study’s findings. The diagnostic/mechanistic viewpoint that CO is responsible for the changes noted in clinical settings of thrombosis and increased carboxyhemoglobin must be reconsidered, with other commonalities besides CO to investigate as the cause of thrombotic morbidity. Further, the CORM-2-generated Ru• could be potentially used to augment clinically available transfusion products, such as fresh frozen plasma or cryoprecipitate prior to administration to bleeding patients. Additionally, systemic administration of CORM-2 could be considered a hemostatic intervention, considering the success seen in rabbit models of coagulopathy [11,12]. However, the knowledge that CORM-2 could serve as a hemostatic agent is not new, and the correct identification of Ru•, not CO, as the molecule responsible for CORM-2-mediated effects may not change clinical/pharmaceutical company interest in development of CORM-2 as a hemostatic agent. Considered as a whole, the findings of our investigation answered the questions posed, and the aforementioned future directions are justified.

In conclusion, this manuscript is a molecular paradigm iconoclasm, serving to replace the putative role of CO derived from CORM-2 as the key modulator of coagulation and fibrinolysis with that of Ru•-mediated effects as the critical mechanism. This new paradigm will impact the underpinning of the hundreds of manuscripts citing the invalidated CO-dependent paradigm generated by the present senior author as displayed in Table 1. However, the association of CO with thromboembolic disease and inflammatory states still exists, but for reasons yet to be identified. Thus, while it may be disappointing to some that a heme-based paradigm may not be entirely responsible for inflammatory thrombotic morbidity, the present work should serve as a rationale to change investigative direction. Lastly, while bittersweet, it is satisfying that the investigator that made the original discovery of the effects of CORM-2 on coagulation and fibrinolysis and proposed an incorrect mechanism had the opportunity to define the true, Ru•-mediated mechanism involved.

## 4. Materials and Methods

### 4.1. Plasma and Chemicals

Pooled normal human plasma that was sodium citrate anticoagulated and maintained at −80 °C was obtained from George King Bio-Medical (Overland Park, KS, USA). This lot of plasma had a prothrombin time of 13.5 s; an activated partial thromboplastin time of 29.6 s; a fibrinogen concentration of 308 mg/mL; Factor II, V, VII, and X activities >94%; Factor VIII and IX activities >92%; and Factor XI and XII activities >121%. These values were within the normal ranges as per the manufacturer. Calcium-free phosphate-buffered saline (PBS), CORM-2, dimethyl sulfoxide (DMSO), and histidine were obtained from Millipore Sigma (Millipore Sigma, Saint Louis, MO, USA). Tissue factor (TF) for activating coagulation was obtained in the form of Pacific Hemostasis™ Prothrombin Time Reagent, Thermo Fisher Scientific, Pittsburgh, PA, USA). Tissue-type plasminogen activator (tPA) was manufactured by Genetech, Inc. (San Francisco, CA, USA). Calcium chloride (200 mM) was obtained from Haemonetics Inc. (Braintree, MA, USA).

### 4.2. Sample Compostion, Incubations, and Coagulation Monitoring

Sample mixtures were placed in a disposable cup in a computer-controlled thrombelastograph^®^ haemostasis system (Model 5000; Haemonetics Inc., Braintree, MA, USA) at 37 °C. The mixture used in the series of experiments was composed of 320 µL of plasma, 10 µL of TF (0.1% final concentration, 10 µL of PBS or tPA (500 U/mL or 0.86 µg/mL final concentration), and lastly 20 µL of calcium chloride. Prior to the placement of plasma samples into the thrombelastographic system, a 1% addition of DMSO or CORM-2 in DMSO (*v*/*v*) was added to 1 mL of plasma for a final concentration of 0 or 100 µM CORM-2. The use of 100 µM CORM-2 is based on a previous concentration–response investigation that demonstrated it to produce maximal change in coagulation and fibrinolytic kinetics [8]. Similarly, the same plasma samples also had a 3% addition of dH_2_O or histidine in dH_2_O for final concentrations of 0, 1, 4, or 8 mM histidine. The order and duration of the addition of vehicles or compounds at room temperature are previously described in the individual series of experiments presented in the Results section. After calcium chloride addition to the plasma samples in the thrombelastographic cup, thrombelastographic parameter values as displayed in Figure 1 were collected. In experiments involving coagulation in the absence of tPA addition, data were collected for 15 min. In experiments utilizing tPA, data were collected until CLT occurred, defined as a return of clot strength to 2 mm, indicative of dissolution of the clot.

### 4.3. Statistical Analyses

Data are presented as mean +SD. All conditions were represented by N = 6 replicates, as this provided a statistical power ≥0.8 with *p* < 0.05 using this methodology to assess differences in thrombelastographic parameters [13,14,15,16,22,23]. A commercially available statistical program was used for unpaired Student’s *t*-test or one-way ANOVA, followed by Holm–Sidak post hoc analyses as appropriate (SigmaStat 3.1; Systat Software, Inc., San Jose, CA, USA). *p* < 0.05 was considered significant. Graphics were generated with commercially available programs; Origen 2024, OrigenLab Corporation, Northampton, MA, USA; and CorelDRAW 2024, Alludo, Ottawa, ON, Canada.

## Figures and Tables

**Figure 1 ijms-26-03567-f001:**
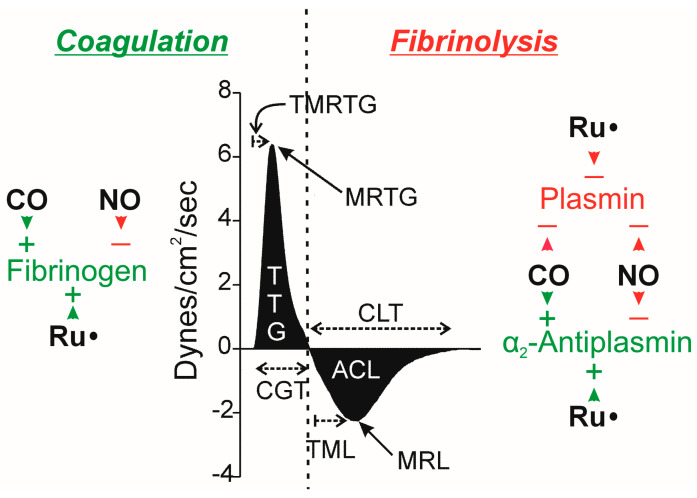
Possible effects of CO, NO, and Ru• on key molecules in the coagulation and fibrinolytic pathways, with relevant thrombelastographic variables. The center portion of the figure displays thrombelastographic output over time during thrombus growth during coagulation followed by data during fibrinolysis, separated with the vertical, dashed line. ACL = area of clot lysis (dynes/cm^2^); CGT = clot growth time (minutes); CLT = clot lysis time (minutes); MRTG = maximum rate of thrombus growth (dynes/cm^2^/second); MRL = maximum rate of clot lysis (dynes/cm^2^/second); TML = time to maximum lysis (minutes), also reported as TMRL, time to maximum rate of lysis—this is the time from maximum clot strength to time of maximum rate of lysis; TMRTG = time to MRTG (minutes); and, TTG = total thrombus generation (dynes/cm^2^). The key molecules involved in the process of coagulation (fibrinogen, green indicating enhancing coagulation) and fibrinolysis (α_2_-antiplasmin, green as it diminishes fibrinolysis; plasmin, red as it enhances fibrinolysis) are depicted within the left and right sides of the dashed line. Green arrowheads with (+) indicate the enhancement of fibrinogen as a substrate or α_2_-antiplasmin enzymatic activity by the indicated molecule; red arrowheads and (−) indicate the degradation of fibrinogen as a substrate or α_2_-antiplasmin or plasmin enzymatic activity. Please refer to the text for the literature supporting these possible molecular interactions.

**Figure 2 ijms-26-03567-f002:**
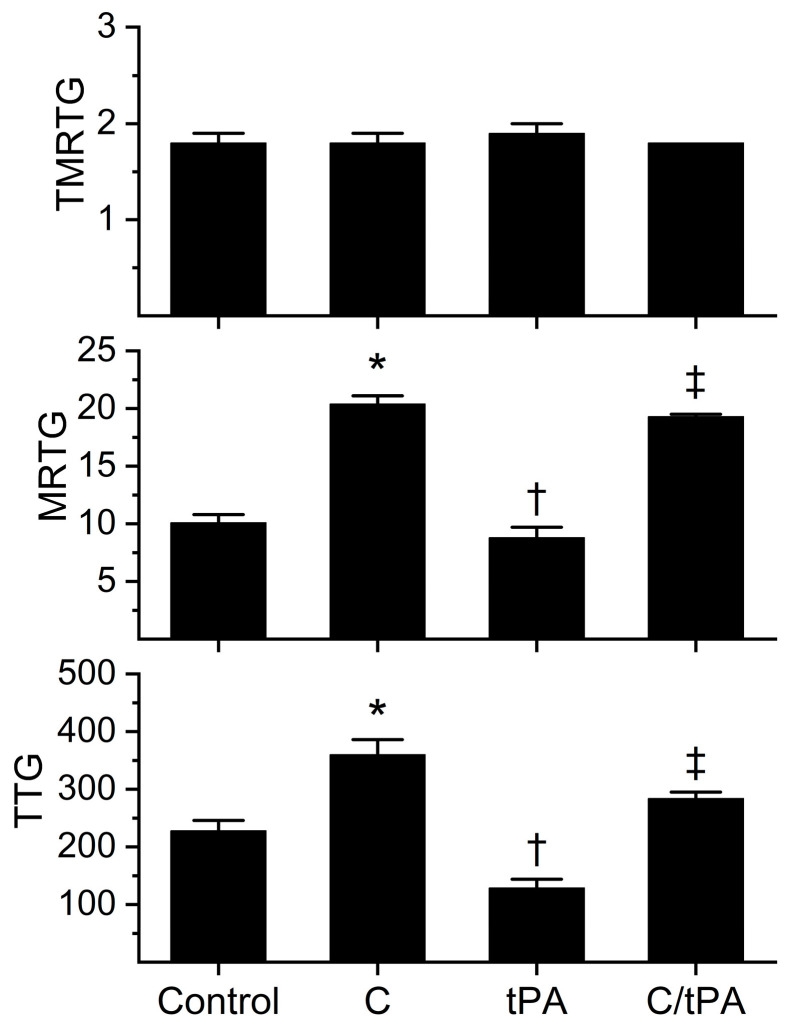
Effects of CORM-2 and tPA on coagulation kinetics in plasma. Control = no additions. C = CORM-2 addition (100 µM final concentration); tPA = tPA addition (500 U/mL final concentration); C/tPA = t PA added to plasma exposed to CORM-2 for 5 min. N = 6 per condition; data analyzed with one-way analysis of variance (ANOVA) with Holm–Sidak post hoc test. * *p* < 0.05 vs. Control, † *p* < 0.05 vs. Control and C, ‡ *p* < 0.05 vs. Control, C, and tPA.

**Figure 3 ijms-26-03567-f003:**
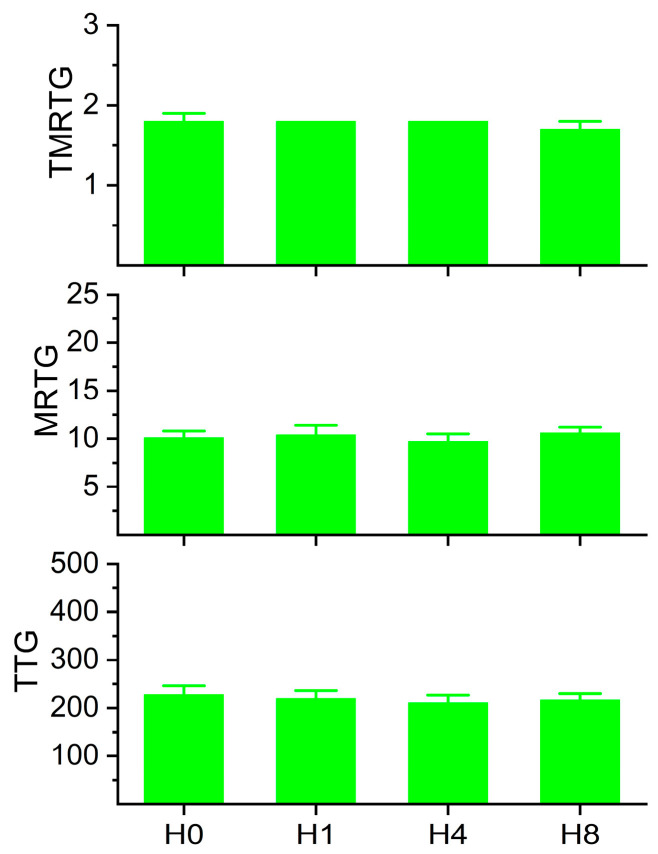
Coagulation kinetic data of plasma exposed to histidine. H0 = no exposure; H1 = exposure to 1 mM histidine; H4 = exposure to 4 mM histidine; H8 = exposure to 8 mM histidine. Analysis with one-way ANOVA demonstrated no significant differences between the conditions. N = 6 per condition.

**Figure 4 ijms-26-03567-f004:**
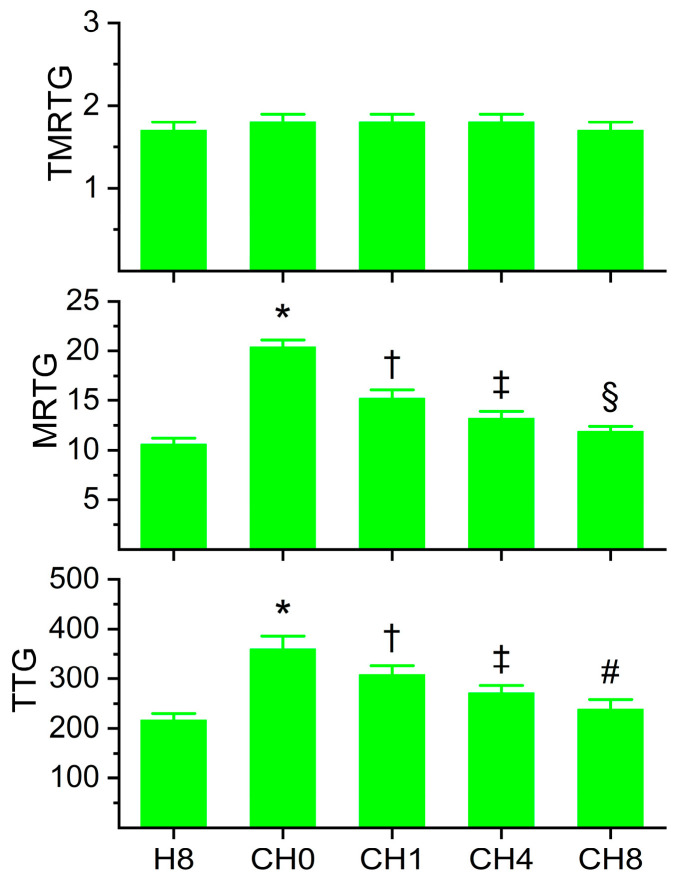
Coagulation kinetic data of plasma exposed to histidine for 5 min before exposure to CORM-2 for 5 min. H8 = exposure to 8 mM histidine; CH0 = no exposure to histidine, exposure to 100 µM CORM-2; CH1 = exposure to 1 mM histidine followed by CORM-2 exposure; CH4 = exposure to 4 mM histidine followed by CORM-2 exposure; CH8 = exposure to 8 mM histidine followed by CORM-2 exposure; and N = 6 per condition; data analyzed with one-way ANOVA with Holm–Sidak post hoc test. * *p* < 0.05 vs. H8; † *p* < 0.05 vs. H8 and CH0; ‡ *p* < 0.05 vs. H8, CH0, and CH1; § *p* < 0.05 vs. H8, CH0, CH1, and CH4; # *p* < 0.05 vs. CH0, CH1, and CH4.

**Figure 5 ijms-26-03567-f005:**
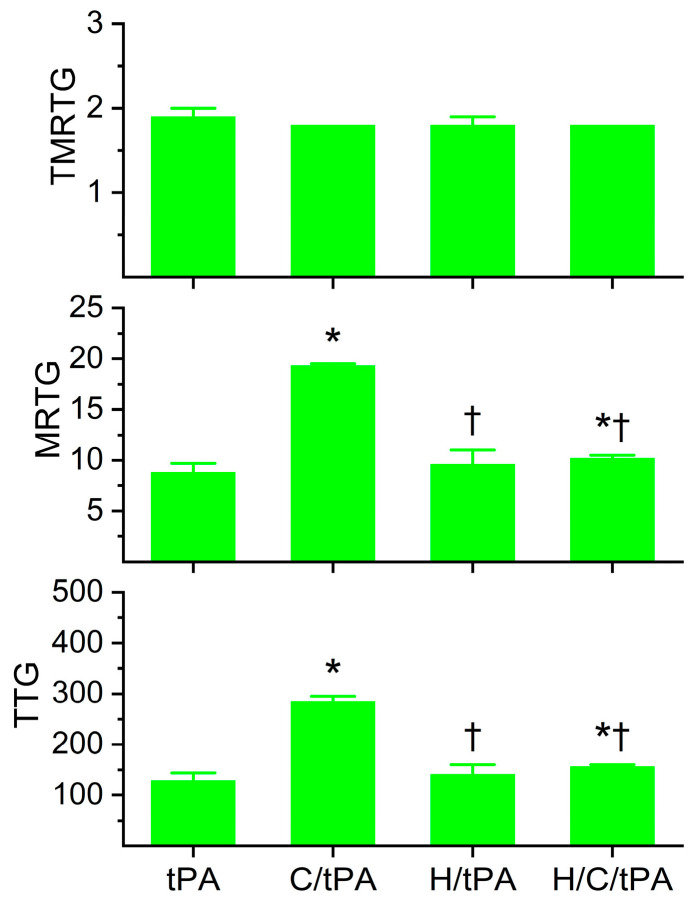
Plasmatic coagulation kinetic data in the presence of tPA, CORM-2, and histidine. tPA = condition with only 500 U/mL tPA added; C/tPA = condition with 100 µM CORM-2 and tPA added; H/tPA = histidine 8 mM and tPA added; H/C/tPA = condition with first histidine added before CORM-2 is added, and tPA. N = 6 per condition; data analyzed with one-way ANOVA with Holm–Sidak post hoc test. * *p* < 0.05 vs. tPA; † *p* < 0.05 vs. C/tPA.

**Figure 6 ijms-26-03567-f006:**
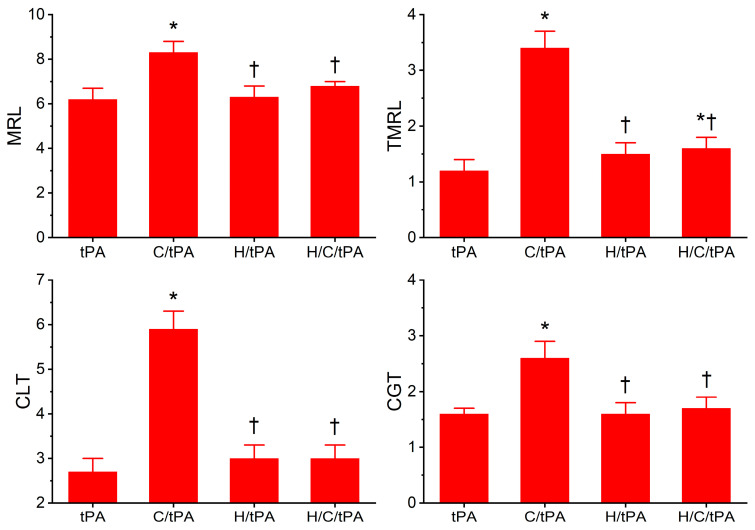
Plasmatic fibrinolytic kinetic data in the presence of tPA, CORM-2, and histidine. tPA = condition with only 500 U/mL tPA added; C/tPA = condition with 100 µM CORM-2 and tPA added; H/tPA = histidine 8 mM and tPA added; H/C/tPA = condition with first histidine added before CORM-2 is added, and tPA. N = 6 per condition; data analyzed with one-way ANOVA with Holm–Sidak post hoc test. * *p* < 0.05 vs. tPA; † *p* < 0.05 vs. C/tPA.

**Figure 7 ijms-26-03567-f007:**
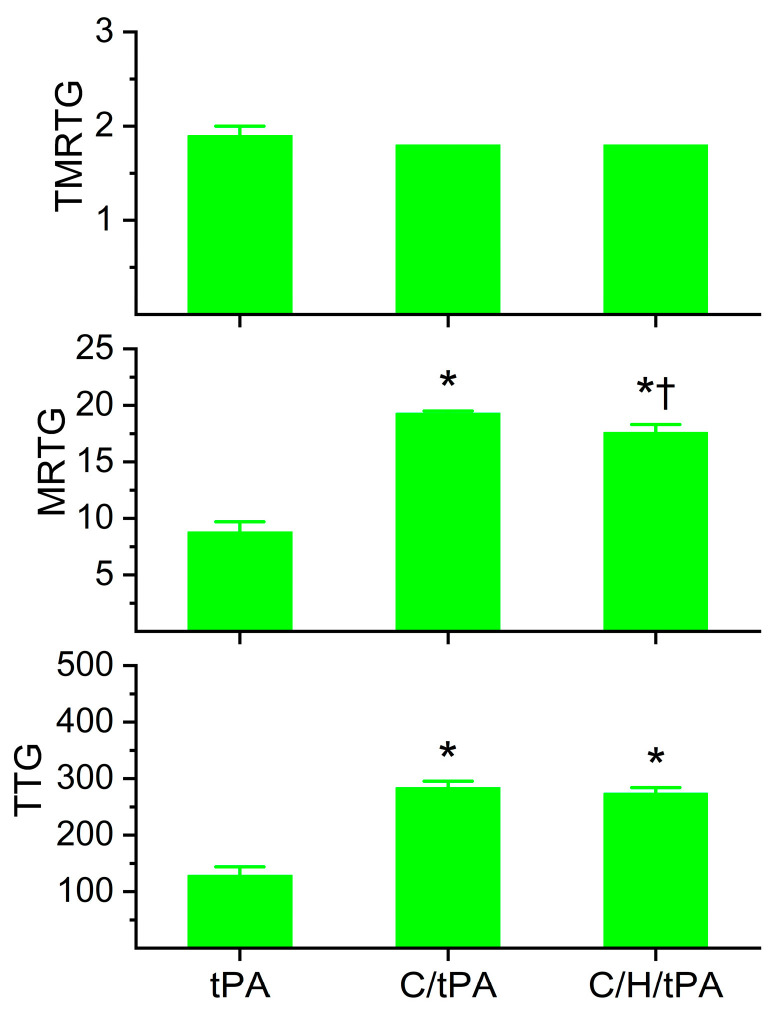
Determination of the reversibility of CORM-2-mediated effects on plasmatic coagulation kinetics following the addition of histidine after CORM-2 addition in the presence of tPA. tPA = condition with only 500 U/mL tPA added; C/tPA = condition with 100 µM CORM-2 and tPA added; C/H/tPA = condition with first CORM-2 added before histidine is added, and tPA. N = 6 per condition; data analyzed with one-way ANOVA with Holm–Sidak post hoc test. * *p* < 0.05 vs. tPA; † *p* < 0.05 vs. C/tPA.

**Figure 8 ijms-26-03567-f008:**
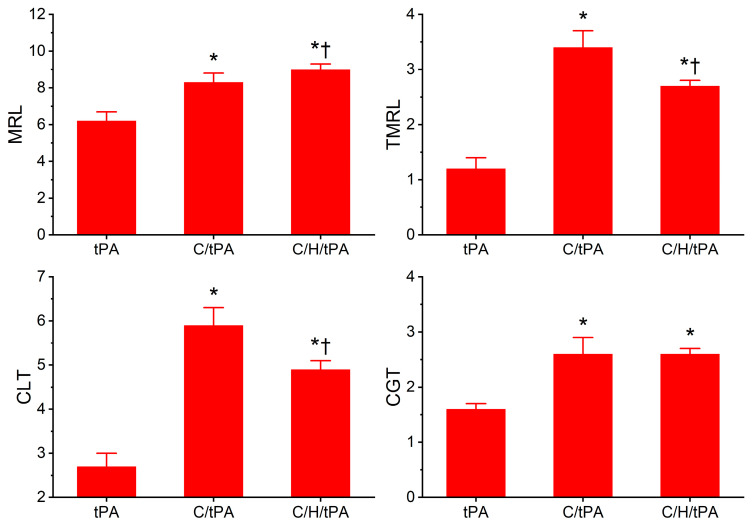
Determination of the reversibility of CORM-2-mediated effects on plasmatic fibrinolytic kinetics following the addition of histidine after CORM-2 addition in the presence of tPA. tPA = condition with only 500 U/mL tPA added; C/tPA = condition with 100 µM CORM-2 and tPA added; C/H/tPA = condition with first CORM-2 added before histidine is added, and tPA. N = 6 per condition; data analyzed with one-way ANOVA with Holm–Sidak post hoc test. * *p* < 0.05 vs. tPA; † *p* < 0.05 vs. C/tPA.

**Table 1 ijms-26-03567-t001:** Manuscripts that involve CORM-2 published by the author and citations of these articles, excluding self-citations as obtained from Google Scholar ^1^.

Manuscripts	Citations ^1^
*Blood Coagul. Fibrinolysis.* **2009**, *20*, 377–380.	30
*Blood Coagul. Fibrinolysis.* **2009**, *20*, 448–455.	21
*Blood Coagul. Fibrinolysis.* **2010**, *21*, 41–45.	6
*Blood Coagul. Fibrinolysis.* **2010**, *21*, 101–105.	12
*Blood Coagul. Fibrinolysis.* **2010**, *21*, 298–299.	2
*Thromb. Res.* **2010**, *126*, 68–73.	5
*Blood Coagul. Fibrinolysis.* **2010**, *21*, 349–353.	11
*Blood Coagul. Fibrinolysis.* **2010**, *21*, 584–587.	10
*Anesth. Analg.* **2010**, *111*, 1347–1352.	4
*J. Trauma.* **2011**, *70*, 939–947	3
*Blood Coagul. Fibrinolysis.* **2011**, *22*, 60–66.	6
*Blood Coagul. Fibrinolysis.* **2011**, *22*, 345–348.	11
*Blood Coagul. Fibrinolysis.* **2011**, *22*, 362–368.	2
*Blood Coagul. Fibrinolysis.* **2011**, *22*, 443–447.	23
*Blood Coagul. Fibrinolysis.* **2011**, *22*, 657–661.	15
*Blood Coagul. Fibrinolysis.* **2011**, *22*, 712–719.	13
*Blood Coagul. Fibrinolysis.* **2011**, *22*, 756–759.	13
*J. Surg. Res.* **2012**, *173*, 232–239.	9
*Thromb. Res.* **2012**, *129*, 793–796.	7
*Blood Coagul. Fibrinolysis.* **2012**, *23*, 104–107.	12
*Blood Coagul. Fibrinolysis.* **2013**, *24*, 273–278.	6
*ASAIO J.* **2013**, *59*, 93–95.	7
*Blood Coagul. Fibrinolysis.* **2013**, *24*, 381–385.	7
*Blood Coagul. Fibrinolysis.* **2013**, *24*, 405–410.	56
*Blood Coagul. Fibrinolysis.* **2013**, *24*, 663–665.	11
*Artif. Organs.* **2013**, *37*, 1008–1014.	10
*Blood Coagul. Fibrinolysis.* **2013**, *24*, 809–813.	13
*Lung Cancer.* **2014**, *83*, 288–291.	5
*Thromb. Res.* **2014**, *133*, 315–321.	25
*Anesth. Analg.* **2014**, *118*, 919–924.	7
*Blood Coagul. Fibrinolysis.* **2014**, *25*, 435–438.	5
*Blood Coagul. Fibrinolysis.* **2014**, *25*, 621–624.	7
*Blood Coagul. Fibrinolysis.* **2014**, *25*, 695–702.	26
*Blood Coagul. Fibrinolysis.* **2014**, *25*, 801–805.	24
*ASAIO J.* **2014**, *60*, 716–721.	9
*J. Thromb. Thrombolysis.* **2015**, *39*, 532–535.	2
*Blood Coagul. Fibrinolysis.* **2015**, *26*, 200–204.	9
*Curr. Neurovasc. Res.* **2015**, *12*, 31–39.	18
*ASAIO J.* **2015**, *61*, 417–423.	6
*CNS Neurol. Disord. Drug Targets.* **2015**, *14*, 1079–1085.	2

^1^ Excluding self-citations: 470 total citations.

**Table 2 ijms-26-03567-t002:** Comparison of fibrinolysis kinetic parameters in the absence or presence of CORM-2.

Parameter	Condition	Value
CGT	tPA	1.6 ± 0.1
	tPA+CORM-2	2.6 ± 0.3 *
TMRL	tPA	1.2 ± 0.2
	tPA+CORM-2	3.4 ± 0.3 *
MRL	tPA	6.2 ± 0.5
	tPA+CORM-2	8.3 ± 0.5 *
CLT	tPA	2.7 ± 0.3
	tPA+CORM-2	5.9 ± 0.4 *

CGT = clot growth time (minutes); TMRL = time to maximum lysis (minutes); MRL = maximum rate of clot lysis (dynes/cm^2^/second); CLT = clot lysis time (minutes). Only the two groups exposed to tPA had fibrinolytic kinetic data to be compared. Data were analyzed with unpaired Student’s *t*-test. * *p* < 0.0001 tPA vs. tPA+CORM-2.

## Data Availability

There are no other data besides that presented in this work.

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
