# Peer review of "Carbon Monoxide or Ruthenium: Will the Real Modulator of Coagulation and Fibrinolysis Please Stand Up!"

_ijms, 2025, doi:10.3390/ijms26083567_

Round 1
Reviewer 1 Report
Comments and Suggestions for Authors
This manuscript presents a compelling and well-executed study that challenges a previously accepted paradigm in the field of hemostasis. The authors investigate the mechanism by which CORM-2 modulates coagulation and fibrinolysis, concluding that the effects previously attributed to CO are instead mediated by a ruthenium-based radical species (Ru•). Through a structured and thoughtful series of thrombelastographic experiments, including the strategic use of histidine to quench Ru• activity, the authors demonstrate that the enhancement of coagulation and the suppression of fibrinolysis are not due to CO release, but rather to interactions involving the Ru• moiety. However, several areas require revision before the manuscript can be recommended for publication.
The central mechanistic claim-that Ru•, and not CO, is responsible for the biological effects observed-would benefit from further clarification. Specifically, the manuscript should include additional explanation or referencing regarding the nature and behavior of Ru• in biological systems, including how it forms, interacts with amino acids, and competes with other potential binding sites in plasma.
The manuscript also includes a lengthy self-referential section summarizing numerous prior studies by the same group, accompanied by an extensive citation table. This section may appear excessive. It would be advisable to summarize the key points more concisely in the text or move the detailed citation list to supplementary material.
The authors do acknowledge certain limitations, such as the exclusive use of in vitro human plasma models and the indirect nature of Ru• detection. However, a more explicit discussion of these limitations in the discussion section (especially regarding generalizability, the absence of cellular or in vivo validation, and the potential non-specific effects of high histidine concentrations) would be beneficial for readers.
Author Response
“This manuscript presents a compelling and well-executed study that challenges a previously accepted paradigm in the field of hemostasis. The authors investigate the mechanism by which CORM-2 modulates coagulation and fibrinolysis, concluding that the effects previously attributed to CO are instead mediated by a ruthenium-based radical species (Ru•). Through a structured and thoughtful series of thrombelastographic experiments, including the strategic use of histidine to quench Ru• activity, the authors demonstrate that the enhancement of coagulation and the suppression of fibrinolysis are not due to CO release, but rather to interactions involving the Ru• moiety. However, several areas require revision before the manuscript can be recommended for publication.”
“The central mechanistic claim-that Ru•, and not CO, is responsible for the biological effects observed-would benefit from further clarification. Specifically, the manuscript should include additional explanation or referencing regarding the nature and behavior of Ru• in biological systems, including how it forms, interacts with amino acids, and competes with other potential binding sites in plasma.”
In section 2.2, first paragraph of Results, we have provided information concerning the use of histidine, the relevant concentrations in previous studies, the competitive histidine and disulfide bridge targets for CORM-2 radicals in plasma, the target histidines on the target molecules (e.g., fibrinogen), and the molar concentrations of all these molecules. The radical formed from CORM-2 is unique, and a variety of works have addressed its binding kinetics which was recently reviewed in one of the senior author’s works [23]. We now include the following sentence in the referenced paragraph to augment this information: “The interested reader will find, in greater detail, information concerning the interactions of CORM-2 derived Ru• with amino acids besides histidine and with disulfide bridges as recently reviewed [23].”
“The manuscript also includes a lengthy self-referential section summarizing numerous prior studies by the same group, accompanied by an extensive citation table. This section may appear excessive. It would be advisable to summarize the key points more concisely in the text or move the detailed citation list to supplementary material.”
We appreciate the reviewer’s comment, but we would ask their indulgence to leave the table as it is. One of the main thrusts of the article (and Special Issue) is to show how paradigm change can affect interpretation of the literature. The inclusion of table 1 seemed the most effective way to do this visually. Further, we did this instead of listing all the articles in the Reference section to avoid unnecessary self-citation. Lastly, using such information in this way was meant to be a model for other authors that may submit to this Special Issue.
“The authors do acknowledge certain limitations, such as the exclusive use of in vitro human plasma models and the indirect nature of Ru• detection. However, a more explicit discussion of these limitations in the discussion section (especially regarding generalizability, the absence of cellular or in vivo validation, and the potential non-specific effects of high histidine concentrations) would be beneficial for readers.”
We appreciate the reviewer’s comments. We never meant for our findings or methods to be made generalizable to cellular systems or clinical investigation/practice, and we have specifically limited ourselves to plasmatic biochemistry. We wanted to ask and answer our questions on the molecular level with validated in vitro methods. We specifically tested the concentrations of histidine to verify that they had no direct effect on plasmatic coagulation, and we did not advocate that these doses of histidine be used in cellular systems.
We include the following statement in our limitations paragraph in Discussion: “A further limitation is the use of our methods beyond the plasmatic biochemical realm. The concentrations of CORM-2, histidine, or any other compounds used in our plasma-based system cannot be adopted into cellular or in vivo systems without requisite concentration-response relationships being defined for the molecular targets of interest and cellular/whole animal toxicity.”
Reviewer 2 Report
Comments and Suggestions for Authors
1. The title is engaging; however, the abstract should more explicitly state the study's novelty compared to prior work clearly. Consider briefly clarifying what previous assumptions were made and how this study definitively challenges them.
2. The introduction provides a good historical context. However, further emphasis on why it is clinically important to clearly distinguish between CO- and Ru-mediated effects in coagulation/fibrinolysis would strengthen the manuscript's clinical relevance.
3. The use of thrombelastography is appropriate and clearly explained. However, consider clearly justifying the rationale behind the chosen concentrations (100 µM CORM-2, 8 mM histidine, 500 U/ml tPA) with relevant literature or preliminary data.
4. Testing an additional ruthenium-based molecule without CO-releasing properties would help confirm that observed effects are strictly Ru-mediated and independent of CO.
5. The authors should consider conducting biochemical validation experiments, such as spectroscopic analyses or mass spectrometry, to confirm direct binding interactions between Ru• and target coagulation/fibrinolysis proteins, specifically fibrinogen, plasmin, or α2-antiplasmin.
6. A detailed dose-response curve of CORM-2 and histidine could further clarify their specific interaction thresholds and provide more precise evidence for the binding affinities and mechanistic interactions being proposed.
7. In the results section, there is occasional speculation (e.g., explanations for increased MRL values). It is advised to clearly label such statements as speculative, or preferably, support them with direct experimental evidence or cite relevant literature clearly.
8. While limitations such as the reversible nature of Ru• binding are briefly discussed, a more comprehensive acknowledgment of potential biological variability, translational limitations, and the need for in vivo validation would strengthen the manuscript’s scientific credibility. Clearly outlining future research directions to explore clinical implications would be valuable.
Author Response
“1. The title is engaging; however, the abstract should more explicitly state the study's novelty compared to prior work clearly. Consider briefly clarifying what previous assumptions were made and how this study definitively challenges them.”
We appreciate the reviewer’s comment and have modified a few key passages within the Abstract to draw attention to the novelty of the present work.
“2. The introduction provides a good historical context. However, further emphasis on why it is clinically important to clearly distinguish between CO- and Ru-mediated effects in coagulation/fibrinolysis would strengthen the manuscript's clinical relevance.”
We appreciate this comment. Clearly, based on the number of citations noted in table 1, the concept that CO is a major player in coagulation was based on studies involving CORM-2. If CO is not critical, and Ru radicals are responsible for the observed in vitro coagulation/fibrinolysis phenomena, then the “house of cards” falls. We have included a paragraph at the end of Introduction and in the Discussion to address the critical clinical significance of our present work.
“3. The use of thrombelastography is appropriate and clearly explained. However, consider clearly justifying the rationale behind the chosen concentrations (100 µM CORM-2, 8 mM histidine, 500 U/ml tPA) with relevant literature or preliminary data.”
We appreciate this comment and will address each compound concentration. First, early work with CORM-2 demonstrated that 100 µM was sufficient to reach optimal changes in clot kinetics after testing a dose range of 0, 25, 50, 100 or 200 μM in plasma [Blood Coagul. Fibrinolysis. 2009, 20, 448-455.]. We now include the following statement in the Methods section: “The use of 100 µM CORM-2 is based on previous concentration-response investigation that demonstrated it to produce maximal change in coagulation and fibrinolytic kinetics [8].”
Regarding the histidine concentrations, these were explained in detail and justified in section 2.2 of Results. First, we had to determine if the maximum amount of histidine we could put into solution with the desired dilutions of plasma by additions affected coagulation. Then, when histidine was found not to affect coagulation directly, we used the concentrations indicated, and eventually used only 8 mM in the fibrinolysis experiments.
As for tPA, the concentration used was justified in section 2.1. We wrote the following passage: “It should be noted that the concentration of tPA was five-fold greater than what has been utilized in the past [8,12,14,15], with the rationale that a greatly increased fibrinolytic stress will result in a more rapid completion of the assays that could better define the effects of CORM-2 and histidine during fibrinolysis.”
We hope that these responses have addressed this issue.
“4. Testing an additional ruthenium-based molecule without CO-releasing properties would help confirm that observed effects are strictly Ru-mediated and independent of CO.”
While we appreciate this suggestion, CORM-2 is a ruthenium dimer, and the radical it forms unique. It is not simply a ruthenium and CO containing compound, and its inactivated form (iCORM-2) that was mentioned in the Introduction has been demonstrated thousands of times not to affect coagulation – and iCORM-2 contains the same two ruthenium atoms, but not in the radical state. We hold that the radical that is transitory, not ruthenium, is responsible for the changes observed in coagulation and fibrinolysis.
The approach of using histidine to quench this CORM-2 generated ruthenium radical was established by several other investigators in years past as cited in the manuscript [20]. It is considered a validated biochemical approach, which is why we choose this method.
“5. The authors should consider conducting biochemical validation experiments, such as spectroscopic analyses or mass spectrometry, to confirm direct binding interactions between Ru• and target coagulation/fibrinolysis proteins, specifically fibrinogen, plasmin, or α2-antiplasmin.”
While we appreciate the reviewer’s suggestion, it is outside the scope of the present work. The investigators using the approach of quenching this radical with histidine in their systems that we quote [20] did not need such methods to define mechanisms. The dozens of articles the senior author published with isolated protein/enzyme exposure to CORM-2 and original work in institutions with collaborators with mass spec allowed us to determine the proteins of interest and potential heme-mediated mechanisms.
To address the reviewer’s comment, we include the following passage in our limitations paragraph: “Another limitation is that further biochemical validation experiments could be performed, such as spectroscopic analyses or mass spectrometry, to confirm direct binding interactions between Ru• and target coagulation/fibrinolysis proteins (fibrinogen, plasmin, or α2-antiplasmin). While such investigation is potentially of interest, it is beyond the scope of the present investigation that had very specific objectives using widely validated methods.”
“6. A detailed dose-response curve of CORM-2 and histidine could further clarify their specific interaction thresholds and provide more precise evidence for the binding affinities and mechanistic interactions being proposed.”
We provided in past literature and the present work the concentration-response data using thrombelastography. If the reviewer is advocating that we produce another manuscript with spectroscopic and mass spec methods with a dose-response, we respectfully would note that it would be far outside the scope of our stated goals. Worse yet, we are publishing whole plasma chemistry results. What the interaction of purified proteins in isolation would be may or may not be representative of what we are documenting in whole plasma.
“7. In the results section, there is occasional speculation (e.g., explanations for increased MRL values). It is advised to clearly label such statements as speculative, or preferably, support them with direct experimental evidence or cite relevant literature clearly.”
We have examined the Results section and tried to identify what the reviewer is noting. On page 7, top paragraph, we found and instance and have added the speculative nature of our explanation. However, we note that we mentioned literature that supported the speculation that contained direct experimental evidence.
For the remainder of Results, we based our comments on the data presented and could not find another clear example of speculation.
“8. While limitations such as the reversible nature of Ru• binding are briefly discussed, a more comprehensive acknowledgment of potential biological variability, translational limitations, and the need for in vivo validation would strengthen the manuscript’s scientific credibility. Clearly outlining future research directions to explore clinical implications would be valuable.”
We appreciate the reviewer’s comments, but there are problems in addressing them. First, it is uncertain what “potential biological variability” is referencing in our experiments demonstrating partial Ru radical binding reversibility. The plasma is standardized, the chemicals standard and pure, and the thrombelastograph optimized. There is no more variability in these experiments than any of the others, and the senior author strives to remove variability that may prevent effective experimentation. Regarding translational limitations, the authors never meant this as a work to be translated into clinical practice. The manuscript was meant to define the mechanism responsible for CORM-2 mediated changes in plasmatic coagulation and fibrinolysis, and anything else is beyond its scope. About in vivo validation, considering the senior author is a clinician, no one would advocate administering human beings high-dose intravenous histidine to see if they could prevent intravenous CORM-2 from changing coagulation/fibrinolysis. The work with rabbit models in the past demonstrated CORM-2 effects in vivo [Blood Coagul Fibrinolysis. 2011 Dec;22(8):756-9.; Blood Coagul Fibrinolysis. 2012 Jan;23(1):104-7.]. The conduct of any further animal or human studies, which would require years and substantial funding, does not seem to be a reasonable request to “strengthen the credibility” of our investigation.
The scientific credibility of our investigation, which the reviewer is questioning, and is a serious matter, is based on nearly two decades of work/publications; based on assessment of how the CORM-2 literature has changed; based on experimental design that is considered state-of-the-art in the coagulation space; based on meticulous execution that resulted in unambiguous results; and, based on analyses with the appropriate statistics by a senior investigator that conceived the original CORM-2/coagulation/fibrinolysis paradigm. In vitro investigations are credible by themselves based on scientific merit.
As for future directions, they are outlined in a new paragraph in the Discussion.
Reviewer 3 Report
Comments and Suggestions for Authors
The paper „Will the Real Modulator of Coagulation and Fibrinolysis Please Stand Up!” is very interesting and touches on an important topic which is CORM-2 as the key modulator of coagulation and fibrinolysis potential therapeutic agents. The discovery of carbon monoxide releasing molecules (CORMs) was very important and is still the subject of scientific research
My commentsThe discussion and conclusions lack references to the potential application of research results in clinical use.
Results; Table 2 – need the explanation of the abbreviations under the table CGT, TMRL, MRL, CLT
Author Response
“The paper “Will the Real Modulator of Coagulation and Fibrinolysis Please Stand Up!” is very interesting and touches on an important topic which is CORM-2 as the key modulator of coagulation and fibrinolysis potential therapeutic agents. The discovery of carbon monoxide releasing molecules (CORMs) was very important and is still the subject of scientific research”
“My comments”
“The discussion and conclusions lack references to the potential application of research results in clinical use.”
We appreciate this comment. The clinical implications of invalidating the rationale for the hundreds of investigations based on CO as the modulator of coagulation and fibrinolysis are noted in table 1, and we have added an additional paragraph in Introduction and Discussion. Further, the senior author noted early on the potential clinical applications of CORM-2 in the cited work that included not just in vitro work but also preclinical rabbit models of coagulopathy [11,12]. We hope these modifications are satisfactory to the reviewer.
“Results; Table 2 – need the explanation of the abbreviations under the table CGT, TMRL, MRL, CLT”
We have made the requested changes.
Round 2
Reviewer 1 Report
Comments and Suggestions for Authors
--
Reviewer 2 Report
Comments and Suggestions for Authors No more comments